# Comparative Cytocompatibility and Mineralization Potential of Bio-C Sealer and TotalFill BC Sealer

**DOI:** 10.3390/ma12193087

**Published:** 2019-09-22

**Authors:** Sergio López-García, Miguel R. Pecci-Lloret, Julia Guerrero-Gironés, María P. Pecci-Lloret, Adrián Lozano, Carmen Llena, Francisco Javier Rodríguez-Lozano, Leopoldo Forner

**Affiliations:** 1Cellular Therapy and Hematopoietic Transplant Unit, Internal Medicine Department, Virgen de la Arrixaca Clinical University Hospital, IMIB-Arrixaca, University of Murcia, 30120 Murcia, Spain; slg4850@gmail.com; 2Department of Genetics and Microbiology, Faculty of Biology, University of Murcia, 30100 Murcia, Spain; 3School of Dentistry, Faculty of Medicine, University of Murcia, 30007 Murcia, Spain; miguelr.pecci@gmail.com (M.R.P.-L.); juliaguerrero1@hotmail.com (J.G.-G.); mpilar.pecci@gmail.com (M.P.P.-L.); 4School of Dentistry, Faculty of Medicine, University of Valencia, 46010 Valencia, Spain; adrianlozano@mac.com (A.L.); llena@uv.es (C.L.); forner@uv.es (L.F.)

**Keywords:** bioceramic, biocompatibility, periodontal ligament stem cells, endodontic sealers, tricalcium silicate

## Abstract

The aim of this study was to investigate the cytocompatibility and mineralization potential of two premixed hydraulic endodontic sealers compared with an epoxy resin-based root canal sealer. The cellular responses and mineralization capacity were studied in human periodontal ligament stem cells (hPDLSCs) that were exposed to premixed hydraulic sealers, Bio-C Sealer (Angelus, Londrína, PR, Brazil), TotalFill BC Sealer (FKG Dentaire SA, La-Chaux-de-fonds, Switzerland) and an epoxy resin-based material, AH Plus (Dentsply De Trey, Konstanz, Germany). Non-exposed cultures served as the control. The endodontic sealers were assessed using scanning electron microscopy (SEM) and energy dispersive X-ray microanalysis (EDX). Statistical analyses were done using Analisis of Variance (ANOVA), with Bonferroni adjusted pairwise comparison (*p* = 0.05). AH Plus reduced cell viability and cell migration, whereas increased cell viability and cell migration were observed in the Bio-C Sealer and the TotalFill BC Sealer (*p* < 0.05). The lowest cell attachment and spreading were observed for all concentrations of AH Plus, whereas the highest were observed for TotalFill BC Sealer. At the end of 21 days, only the Bio-C Sealer and the TotalFill BC Sealer supported matrix mineralization (*p* < 0.05). Additionally, SEM-EDX revealed high content of calcium, oxygen, and silicon in the Bio-C Sealer and the TotalFill BC Sealer. Based on the results from this study, Bio-C Sealer and TotalFill BC Sealer demonstrated better cytocompatibility in terms of cell viability, migration, cell morphology, cell attachment, and mineralization capacity than AH Plus.

## 1. Introduction

In endodontic research, it is very important to know the toxicity or biocompatibility of new materials prior to their clinical application, as the compounds may potentially damage surrounding tissues [1,2]. However, new bioactive materials have been developed for use in dentistry, with a variety of clinical applications, ranging from vital pulp therapy to root canal filling, root-end filling, apexification and perforation repair [3,4,5]. In all applications, these materials are placed in contact with vital tissues, which may be the dental pulp or the periodontium, and facilitate biomineralization [6], the process by which a living organism synthesizes mineral substance. 

In this context, stem cells from dental tissues and their interaction with the endodontic materials have been considered an important event that favors biomineralization [7]. Thus, stem cells from periodontal tissues (hPDLSCs) have been previously used as cell models for in vitro cytotoxic studies involving endodontic sealers, because these cells may be in direct contact with sealer extrusions [8,9,10,11]. These cells have colony forming ability, spindle-like cell morphology, specific cell-surface marker expression, multipotent differentiation, and immunomodulatory functions [11].

Bioceramic or hydraulic materials are promising bioactive candidates for hard tissue repair owing to their excellent physicochemical and biological properties [12]. These materials are composed of silicates (dicalcium/tricalcium) or tricalcium aluminate [13], and capable of producing hydroxyapatite when incorporated with calcium and silicon, showing functional bonding with dentin [14].

The Bio-C Sealer (Angelus, Londrína, PR, Brazil) is a new, premixed bioceramic sealer developed for permanent filling and sealing during root canal treatment. Bio-C Sealer is available in a single syringe, composed of calcium silicates, calcium aluminate, calcium oxide, zirconium oxide, iron oxide, silicon dioxide, and dispersing agents. According to the manufacturer, its bioactivity is attributed to the release of calcium ions that stimulate the formation of mineralized tissue. However, to date, few studies have evaluated its effects on periapical tissues and related cells. TotalFill BC Sealer (FKG Dentaire SA, La-Chaux-de-fonds, Switzerland) is another calcium silicate bioceramic based sealer that has shown good physical and biological properties and has the ability to release calcium ions. 

The present research aimed to investigate the cytocompatibility and mineralization potential of two premixed hydraulic endodontic sealers compared with an epoxy resin-based root canal sealer (AH Plus). The null hypothesis was that there is no difference between the tested materials in their mineralization potential or cytocompatibility to human periodontal ligament stem cells.

## 2. Material and Methods

### 2.1. Material Extracts

The following endodontic sealers were used: Bio-C Sealer (Angelus, Londrína, PR, Brazil), TotalFill BC Sealer (FKG Dentaire SA, La-Chaux-de-fonds, Switzerland) and AH Plus (Dentsply De Trey, Konstanz, Germany). 

Under sterile conditions in a laminar flow hood, the sealers were mixed according to the manufacturers’ instructions. The material discs were molded in a sterile, cylindrical, polyethylene tube (diameter: 5 mm; height: 2 mm) and stored in a dark container at 37 °C for 48 h to allow complete setting (n = 30). After this period, sample disks were stored in the culture medium (DMEM, Sigma-Aldrich Química SL, Madrid, España) for 24 h at 37 °C, 5% CO_2_ and a humid atmosphere. The preparation of material extracts was performed according to previous studies and the International Organization for Standardization (ISO) guideline 10993-12 [15]. The ratio of the specimen surface area was 1.5 cm^2^/mL (ISO 10993-5) [15]. Sealer extracts were prepared, filtered and diluted. Three different dilutions (undiluted, 1/2, 1/4) were used.

### 2.2. Cell Cultures

This study used stem cells from human periodontal tissues (hPDLSCs) that had been previously characterized by our group [15]. These cells were obtained from impacted third molars (n = 12) from 10 healthy subjects. Donors gave written, informed consent according to the guidelines of the Ethics Committee of our institution (2017-3-8-HCVA). The hPDL was scraped from the middle, third region of the root surface. After extraction, hPDL was washed with Ca^2+^/Mg^2+^ -free Hank’s balance salt solution (Gibco, Gaithersburg, MD, USA) and subjected to collagenase-A digestion (3 mg/mL) (Sigma–Aldrich, St. Louis, MO, USA) for 1 h at 37 °C. hPDLSCs were seeded in 75 cm^2^ flasks (Corning Incorporated, Corning, NY, USA) and cultured in expansion medium composed of α-Minimum Essential Media (Invitrogen, Grand Island, NY, USA), 10% fetal bovine serum (FBS) (Invitrogen, Eugene, OR, USA), 100 IU/mL penicillin, and 100 μg/mL streptomycin (Sigma, St. Louis, MO, USA). The culture media was replaced every 3 days. Cell culture was carried out at 37 °C within a humidified 5% CO_2_ incubator. hPDLSCs between passages 3 to 5 were used for all experiments.

### 2.3. Cell Viability Assay

Cytotoxicity was determined using a 3-(4,5-dimethylthiazol-2-yl)-2,5-diphenyltetrazolium bromide (MTT) (Sigma-Aldrich, St. Louis, MO, USA) assay [16]. Briefly, hPDLSCs (3.0 × 10^3^ cells/well) were seeded in 96-well plates and cultured for 24, 48 and 72 h in the presence of material extracts. The samples were incubated with 1 mg/mL of MTT for 4 h at the indicated time points. Then, the medium containing MTT solution was removed, and 0.2 mL of dimethyl sulfoxide (DMSO, Sigma-Aldrich Química SL, Madrid, Spain) was added to each well to dissolve the MTT formazan. The dissolved formazan solution of each well was transferred to a 96-well plate and measured at 570 nm in microplate reader (BioTek, Winooski, VT, USA). Each condition was analyzed in triplicates and the data obtained for each group were expressed as mean ± standard deviation.

### 2.4. Cell Migration

To evaluate cell migration, a scratch wound test was performed, in which a straight scratch was made at the central region of the wells of 24-well plates using a 100 μL pipette tip, causing a rupture between the cells and developing mechanical injury, as previously described [16,17]. The culture medium was then replaced with the sealer extracts at different concentrations, without FBS, in order to prevent cell proliferation. Scratch images were obtained at time 0, and a reference point was determined outside the plate using ImageJ (1.51h version, National Institutes of Health, Bethesda, MD, USA). Subsequent percentage cell migration was quantified from the images captured after 24 h of exposure to the sealer extracts. Migration distances were analyzed separately during periods 0–24 h (migration during first 24 h period), 24–48 h (during second 24 h period), and 48–72 h (during third 24 h period).

### 2.5. Cell Morphology

To analyze cytoskeleton changes, hPDLSCs were seeded directly on glass coverslips at a low density and cultured in culture medium containing undiluted extracts of the different endodontic sealers. After 72 h, the cells were fixed with 4% paraformaldehyde in PBS for 30 min and permeabilized with 0.25% Triton-X-100 (Sigma–Aldrich, St. Louis, MO, USA) dissolved in PBS for 20 min. After the cells were blocked with 2% bovine serum albumin, and the actin filaments were stained with CruzFluor594-conjugated phalloidin (Santa Cruz Biotechnology, Dallas, TX, USA), followed by fluorescent-labeled secondary anti-mouse antibody (Invitrogen, Carlsbad, CA) incubation at room temperature for 1 h. DAPI (4, 6 diamidino-2-phenylindole, Sigma-Aldrich, Madrid, Spain) was used to stain cell nuclei (1:1000 dilution). The cells were examined under epifluorescence using an AxioImager M2 Zeiss light microscope (Carl Zeiss, Oberkochen, Germany), equipped with a digital camera, AxioCam MRM (Carl Zeiss).

### 2.6. Cell Attachment and Surface Morphology

Sealers were made with discs (5 mm diameter and 2 mm thick) fabricated from cylindrical polyethylene tube in sterile conditions and were subdivided into three groups (n = 5). After transferring the disc into a 48-well plate, hPDLSCs (5 × 10^4^ cells/mL) were directly seeded onto each set disc and incubated in 5% CO_2_ at 37 °C for 72 h. The cells were fixed using 4% glutaraldehyde in PBS for 4 h at 4 °C. Subsequently, the samples were subjected to dehydration, air-drying, and 100-nm-thick gold/palladium coating. Finally, discs were visualized using scanning electronic microscopy (SEM) (SEM; JSM-635F, JEOL, Tokyo, Japan), and the magnifications used were 100× and 300×. Then, the specimens were coated with carbon and the changes in the surface morphology and composition were studied using Scanning electron microscopy (SEM; JSM-635F, JEOL, Tokyo, Japan) equipped with energy dispersive X-ray spectroscopy (EDX). An accelerating voltage of 20 kV was used for topographic and compositional evaluations. After obtaining the spectrum, the chemical elements were quantified (atomic% and weight%) [18].

### 2.7. Alizarin Red (AR) Assay

Calcium mineralization was measured using the AR assay as previously described [19]. In this experiment, six discs were immersed in culture medium for 24 h. hPDLSCs (2 × 10^5^/well) were seeded in a 24-well plate, and after 90% confluence, the cells were treated with extracts of endodontic sealers (dilution 1:1) for 21 days, the medium being changed every 3 days. A negative control (without extracts) and a positive control for osteogenic differentiation, using OsteoDiff media (Miltenyi Biotech), were carried out. Next, the culture medium was removed from each well and cells were washed 3 times with 1× PBS and then fixed with 70% ethanol for 1 h at 4 °C. The fixative was then removed, and specimens were washed with deionized water, cleaned and stained with 2% (w/v) Alizarin Red S (Sigma AB, Malmö, Sweden) (pH = 4.1–4.2) at room temperature. The absorbance readings were observed at 550 nm with a SPECTRAMax 340^®^ microplate reader (SPECTRAMax 340, Molecular Devices, Sunnyvale, CA, USA).

### 2.8. Statistical Analysis

The data distribution was normal in all the experiments and hence parametric tests were employed for statistical analysis. For the Alizarin Red assay, one-way ANOVA with Bonferroni adjusted pairwise comparison was performed (*p* = 0.05). For all other experiments, data analysis was done using two-way ANOVA with Bonferroni adjusted pairwise comparison for each time point.

## 3. Results

### 3.1. MTT Assay

The metabolic activity of hPDLSCs in contact with the endodontic sealers extracts varied with time and the materials used (Figure 1). TotalFill BC Sealer and Bio-C Sealer did not affect cell viability in any sample extract in the first 24 h, while AH Plus showed significant cytotoxicity in all dilutions at the same period. Notably, significant differences in cell viability were observed after incubation with the 1:2 of all materials compared to the control group at 48 h. At 72 h, cell viability was significantly higher in the dilutions of 1:2 and 1:4 of TotalFill BC Sealer compared to that of the medium only (** *p* < 0.01; *** *p* < 0.001). However, Bio-C Sealer at 1:4 cell viability was similar in comparison to the control group. Both TotalFill BC Sealer and Bio-C Sealer were significantly less cytotoxic than AH Plus in all dilutions.

### 3.2. Cell Migration

Cell migration rates in undiluted Bio-C Sealer and TotalFill BC Sealer groups were slightly lower in comparison to the control group (* *p* < 0.05; ** *p* < 0.001) (Figure 2). In the undiluted TotalFill BC Sealer, only at 24 h, in the undiluted group, were significant differences not found. Meanwhile, statistical differences were revealed in the 1:2 and 1:4 dilutions when compared with the control group wound closure. On the other hand, at all times and all dilutions, the AH Plus group showed statistical differences (*** *p* < 0.001) and were unable to heal the wound when compared with the control wound closure rates. These results indicate that both bioceramic sealers had similar migration values.

### 3.3. Cell Morphology

Seventy-two hours following exposure of the cell cultures to the extracts of different endodontic sealers, adhered and widely spread cells were observed in Bio-C Sealer and TotalFill BC Sealer groups (Figure 3). However, cells exposed to AH Plus showed typical aspects of cell death with pyknotic nuclei, predominantly rounded shape, reduced spreading, shrinkage, and few cytoplasmic extensions. 

### 3.4. Cell Attachment on Materials and Characterization of Set Materials 

hPDLSCs seeded on TotalFill BC Sealer disks were well spread, with a predominant fibroblastic shape with multiple cytoplasmic extensions. In the case of Bio-C Sealer, less elongated and spindle-shaped cells were found on the surface, and finally, on the AH Plus, there was a significant reduction in density and spreading (Figure 4). 

The endodontic sealers tested showed different surface morphologies in the SEM analysis (Figure 5). Bio-C Sealer showed irregular crystalline structures on the surface, whereas fewer particles were detected in TotalFill BC sealer. In contrast, the surface of AH Plus was homogeneous with few particles. The EDX analysis gave different results for endodontic sealers in terms of percentage of weight. The presence of carbon, oxygen, silicon, calcium and zirconium was detected in the three materials, but in different percentages. Aluminum and magnesium were found in Bio-C Sealer and tungsten in AH Plus. 

### 3.5. Alizarin Red Assay

At the end of 21 days, the Alizarin Red S staining for calcium deposits revealed significant differences in osteogenic potential among the endodontic sealers examined (Figure 6). TotalFill BC Sealer and Bio-C Sealer groups exhibited higher mineralized matrix formation than negative control (* *p* < 0.05). However, for the AH Plus group, no mineralization was detected.

## 4. Discussion

In this study, the cytocompatibility and mineralization potential of two premixed, hydraulic endodontic sealers compared with an epoxy resin-based root canal sealer were evaluated using human periodontal ligament stem cell cultures. Bio-C Sealer and TotalFill BC Sealer are bioactive in nature; i.e., they induce mineralization when in contact with biological tissues [20]. AH Plus was used as a control because it is one of the most commonly used and investigated root filling materials [11,21,22]. 

Several methods have been described to evaluate the biological effects of the endodontic sealers in vitro [23,24]. To test the cytotoxicity, the MTT assay was used due to its simplicity, reliability, accuracy and time-saving attributes [25]. In this study, TotalFill BC Sealer and Bio-C-Sealer showed higher cell viability than AH Plus. In agreement with our results, TotalFill BC Sealer has also demonstrated cytocompatibility in human periodontal ligaments cells (hPDL) [25] and human osteoblast-like cells [26]. The elution of calcium ions by the bioceramic materials could have resulted in cell viability of hPDLSCs [27]. In addition, it may also be possible that the proprietary additives in the liquid may play a role in the enhanced biocompatibility of this material. On the other hand, the cytotoxicity of AH Plus observed in the present study agrees with previous reports [28,29], and may be related to the release of formaldehyde from amines added to accelerate the polymerization of epoxy resin and to the presence of bisphenol-A, known for its toxicity [21]. 

In regenerative medicine, cell migration and adhesion of stem cells are key phenomena during apical healing, because of their participation on local tissue repairs in a variety of physiological and pathological conditions [30]. In this study, it was verified that Bio-C Sealer and TotalFill BC Sealer supported cell migration; this finding could be related to the positive effects of calcium ions on cell migration [28]. Exposure of cultures to AH Plus extracts reduced cell migration even at low concentrations. In epoxy resin sealers, the presence of cytotoxic by-products from polymerization reactions of the epoxy resin may have adversely affected cell migration [31]. 

It is well known that the nature of the initial interaction between biomaterials and cells can influence cell function and their ability to produce an osteoid matrix and is thus a good predictor of their cytocompatibility [32]. Immunofluorescence and SEM assays showed the death of all hPDLSCs on the surface of AH-Plus and in the presence of undiluted extracts. The reduction of cell attachment and the change in classical cell morphology can thus be directly associated with the epoxy resin treatment [25,33]. Conversely, hydraulic cements exhibited cells that were well attached and proliferated on the set surface, which means that they would be more advantageous to cell biocompatibility even in direct contact conditions than epoxy resin-based sealers. Thus, it may be speculated that the biological properties and differentiation of the materials are influenced by their composition [34,35].

SEM-EDX revealed high content of calcium, oxygen, and silicon in Bio-C Sealer and TotalFill BC Sealer. This finding suggests that these materials would probably favor bioactivity or biomineralization and would be expected to interact with the apical tissues [36]. Other root canal sealers containing calcium silicates in similar amounts have also demonstrated bioactivity [12,37]. In our study, all materials contained zirconium. Zirconium oxide is an alternative radiopacifier, which has been recently manufactured and used to limit the content of heavy metals and substitute bismuth oxide in calcium silicate-based materials [26]. Zirconium oxide has become popular due to acceptable physicochemical properties, adequate radiopacity in accordance with the ISO 6876/2001 specifications and lack of interference with the hydration of calcium silicate-based materials [38].

Finally, Alizarin Red assay was performed to evaluate the osteogenic potential of the sealers. The ability of endodontic sealers to promote cell differentiation and mineralization is important for recovery of lost structures, as in apical periodontitis [13,19]. Ca^2+^ ion release, and high pH have been considered responsible for the beneficial effects of calcium silicate-based sealers [13,15,19]. Particularly, the release of calcium into the tissues may enhance biomineralization by promoting activation of calcium-dependent adenosine, cell migration and calcium crystals deposition, which activate the mineralization [3,4,39]. The current report is the first to show that Bio-C Sealer induced mineralization. Unsurprisingly, Bio-C Sealer and TotalFill BC Sealer were significantly higher than AH Plus. These results were in line with those of a previous study that showed that EndoSequence BC, BioRoot RCS, and Endoseal MTA showed a significant increase in mineralized nodule formation compared to the AH Plus group [11]. In addition, a recent study with endodontic sealers showed that TotalFill BC Sealer induced alkaline phosphatase activity and strong mineralization capacity [26]. Although these results are promising, further investigations are required to determine its effects on periapical tissue repair.

## 5. Conclusions

Taken together, Bio-C Sealer and TotalFill BC Sealer demonstrated better cytocompatibility in terms of cell viability, migration, cell morphology, cell attachment, and mineralization capacity than AH Plus. It should be noted that material composition plays an important role in their biological properties.

## Figures and Tables

**Figure 1 materials-12-03087-f001:**
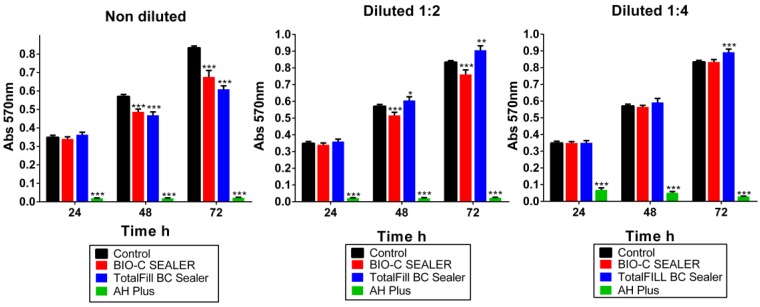
Human periodontal ligament stem cells viability after stimulation with Bio-C Sealer, TotalFill BC Sealer, and AH Plus, as determined by an MTT assay. (* *p* < 0.05; ** *p* < 0.01; *** *p* < 0.001, respectively).

**Figure 2 materials-12-03087-f002:**
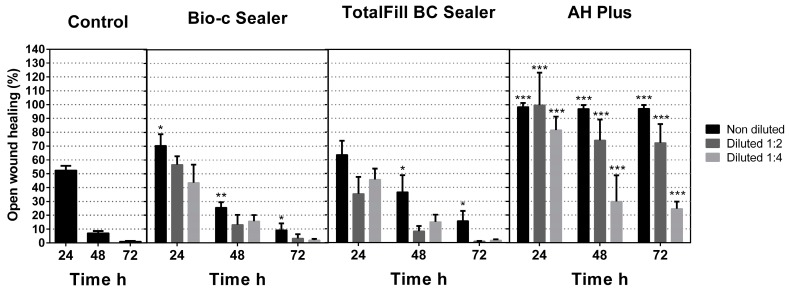
Cell migration of hPDLSCs by scratch wound test using extraction media derived from the set sealers at 24 h for up to 72 h. Cell migration is represented as the percentage of the open wound area for each condition compared with the control. Significant differences are indicated as * *p* < 0.05; ** *p* < 0.01; *** *p* < 0.001, respectively.

**Figure 3 materials-12-03087-f003:**
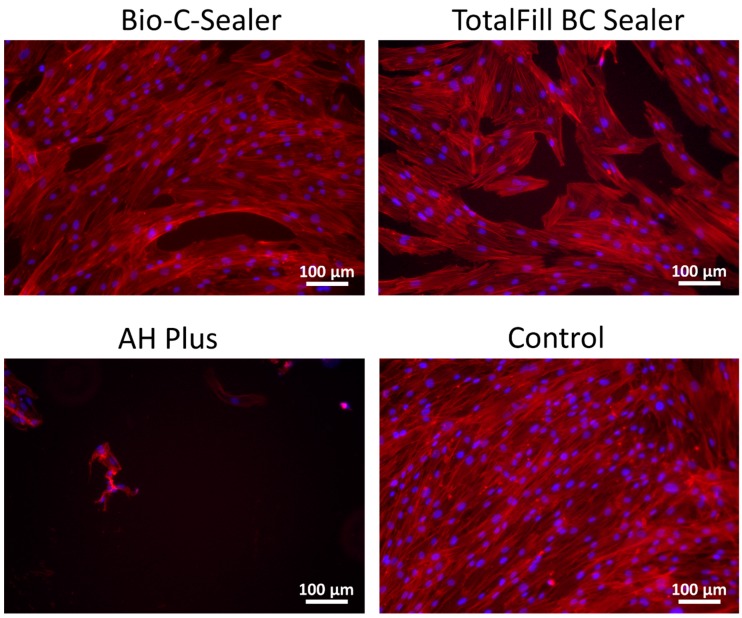
Morphological aspects of hPDLSCs cell cultures exposed to the undiluted extracts of Bio-C Sealer, TotalFill BC Sealer and AH Plus in culture medium. Blue fluorescence (DAPI) indicates cell nuclei, and red fluorescence (phalloidin), the actin cytoskeleton. Scale bar = 100 μm.

**Figure 4 materials-12-03087-f004:**
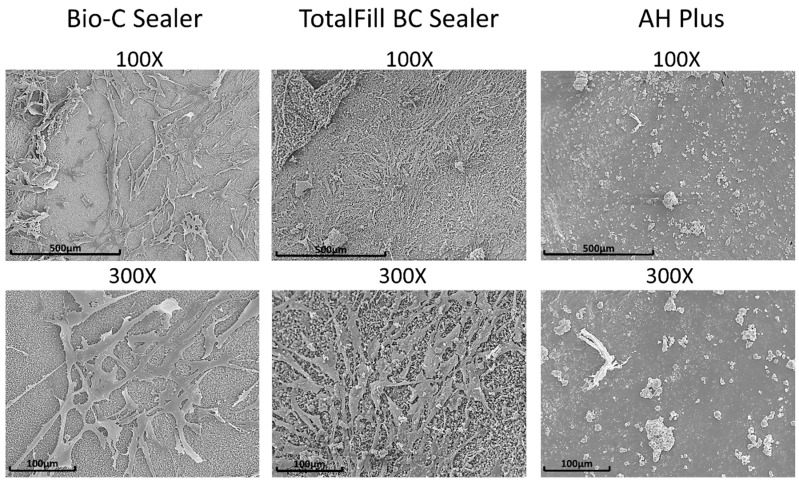
The morphology of hPDLSCs attached on the surface of Bio-C Sealer, TotalFill BC Sealer and AH Plus after culture for 3 days at a magnification of 100× and 300×. Scale bar: 500 μm and 100 μm.

**Figure 5 materials-12-03087-f005:**
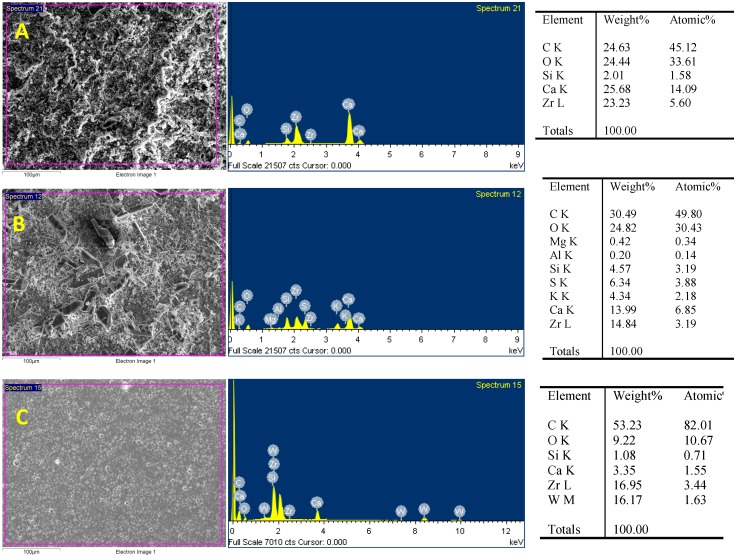
Surface properties and composition of Bio-C Sealer (**A**) and TotalFill BC Sealer (**B**) and AH Plus (**C**) under a scanning electron microscope with energy-dispersive X-ray analysis (SEM-EDX).

**Figure 6 materials-12-03087-f006:**
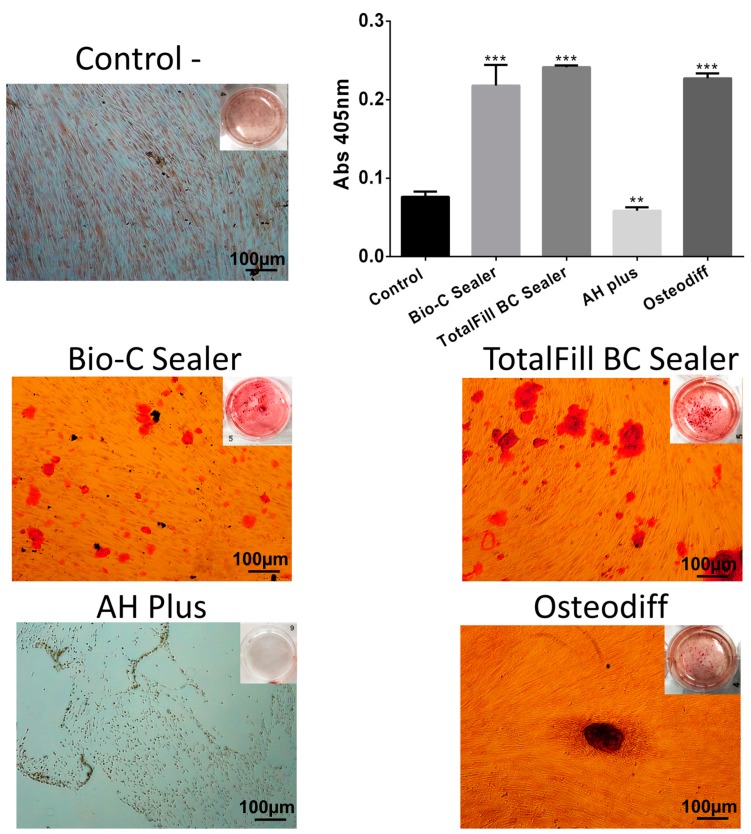
Induction of mineralization and Alizarin Red S staining. The effects of sealers on mineralization potential after 21 day of exposition. Data are presented as the mean ± standard deviation percentage of staining compared with the control (* *p* < 0.05; ** *p* < 0.01; *** *p* < 0.001). Scale bar = 100 μm.

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
