# Peer review of "Comparative Cytocompatibility and Mineralization Potential of Bio-C Sealer and TotalFill BC Sealer"

_materials, 2019, doi:10.3390/ma12193087_

Round 1

Reviewer 1 Report

The authors have shown an interesting finding on cell viability and mineralization of hydraulic vs epoxy resin based sealers. They have performed necessary experiments to conclude the hydraulic sealers are better than epoxy based sealers with respect to cell viability and mineralization. Few suggestions to improve the manuscript before it gets accepted for publication in Materials:

1) The abstract should be rewritten so that it is helpful for the readers to understand. check for abbreviations such as what is hPDLSCs it's not clear until someone reads the introduction.  

2) Although the cell line has been characterized previously, the authors should include some supplementary figures (staining, FACS data) on the characterization of the cells.

3) Add few sentences in the method section  how the cells were derived? If it is from human tissue mention the protocol and whether it was approved by the institution?

4) Cell viability data looks good but not enough. Figure 3 is not representative of the rest of the cell data. The authors should perform more cell viability assays such as staining with caspase, Ki67 and live & dead assays.

5) The affect of these sealers on the cells is not clearly understood with just mineralization and cell viability. The authors should perform additional experiments such as alkaline phosphatase staining, gene expression for various osteogenic markers.

6) Overall a nice study and should be interesting to the readers. 

Author Response

First of all we would like to thank the editor and the referees for the interest they have shown in our article entitled "Comparative cytocompatibility and mineralization potential of Bio-C Sealer and TotalFill BC Sealer." (materials-594755).

We wish to thank the editor and the referees for their constructive comments and suggestions. Please find attached the new version which it has been reviewed in line with your comments and those of the referees.

As it is mentioned in your letter, changes are highlighted in the new manuscript, using the "track changes" mode in MS Word format.

Reply to Reviewer #1:

Concern of the reviewer: 1) The abstract should be rewritten so that it is helpful for the readers to understand. Check for abbreviations such as what is hPDLSCs it's not clear until someone reads the introduction.

Our response: Thank you very much. Change performed

Concern of the reviewer: Although the cell line has been characterized previously, the authors should include some supplementary figures (staining, FACS data) on the characterization of the cells.

Our response: Thank you for your comments. Our apologies, due to the short time for the review (only 7 days), we have not been able to add more experiments. But as you can see in other papers from our group (references 14, 15, 16, 17, 18 and 22), we always perform cell characterization.

Concern of the reviewer: Add few sentences in the method section how the cells were derived? If it is from human tissue mention the protocol and whether it was approved by the institution?.

Our response: Thank you for your comments. We added more information in the Material and Methods section.

Concern of the reviewer: Cell viability data looks good but not enough. Figure 3 is not representative of the rest of the cell data. The authors should perform more cell viability assays such as staining with caspase, Ki67 and live & dead assays.

Our response: Thank you very much. It is well known that the nature of initial interaction between cells and biomaterials can influence cell function and their ability to produce an osteoid matrix, and is a good predictor of their biocompatibility (Kumari et al. 2002) (Added in Discussion Section). In this study, the morphological characteristics of cells exposed to Bio-C Sealer and TotalFill were similar to those of the control group (cells grown in the absence of sealer extracts), exhibiting high cell density and ability to spread. So, we considered necessary this aspect. On the other hand, due to the short time for the review (only seven days), we have not been able to add more experiments.

Concern of the reviewer: The affect of these sealers on the cells is not clearly understood with just mineralization and cell viability. The authors should perform additional experiments such as alkaline phosphatase staining, gene expression for various osteogenic markers.

Our response: Thank you for your comments. In this study, we analyzed of cell viability, migration, cell morphology, cell attachment, and mineralization capacity, not only cell viability and mineralization. In addition, the endodontic sealers were assessed using scanning electron microscopy (SEM) and energy dispersive X ray microanalysis (EDX). Our apologizes, due to the short time for the review (only 7 days), we have not been able to add more experiments.

Concern of the reviewer: Overall a nice study and should be interesting to the readers.

Our response: Thank you very much. We are so happy with this comment.

Reviewer 2 Report

Manuscript ID: materials-594755

“Comparative cytocompatibility and mineralization potential of Bio-C Sealer and TotalFill BC Sealer.”

The aim of this study was to compare two calcium silicate-based sealers with a conventional epoxy resin-based sealer. The main parameter of the study was the effect of the sealers on human periodontal ligament stem cells (hPDLSC).

Introduction

Unfortunately, the current literature on the subject has not been thoroughly reviewed. Some studies, that have investigated the effect of calcium silicate-based sealers on PDL cells, are not cited.

The sentence “Classically, ideal endodontic sealers should be bio-inert, i.e., they do not interact with the host tissues to form a mineralized interface [2].“ doesn't make any sense to me. Why shouldn't sealers induce mineralization?

Material and Methods

In this study, the cells were only contacted with eluates from hard-set sealer. However, this does not correspond to a clinical scenario where freshly mixed, unset sealer can be pressed over the apex during root canal filling and come into direct contact with the cells ("puff"). This may lead to different results than indicated here.

Why were these dilutions chosen? Why no other dilutions? Could this have led to different results? This point needs to be discussed.

The study also investigated the mineralization performance of the cells. Why were osteoblasts, which are responsible for bone mineralisation, not tested?

In the present study it set AH Plus was cytotoxic to the hPDLSCs. This is in contrast to other studies where it was shown that after setting, AH Plus was no longer cytotoxic [Eldeniz et al. 2007, Zhou et al. 2015, Silva et al. 2016, Jung et al. 2019]. This point needs to be discussed.

Page 3, line 95: what does "OD" mean? Please expand abbreviations.

Please, “Carl Zeiss“ without “Inc.“, TotalFill without “®“

Discussion

The discussion suddenly breaks off and came to a sudden end. The methods are not clearly discussed and current literature is not cited. Please, avoid the use of the possessive pronoun “we“.

Figures

Fig. 1: The image resolution in Fig. 1 is very low. The graphs are therefore difficult to decipher.

Fig. 2: Why are the photos of the scratch wounds and the graphs in an image? That makes little sense to me. The units of the y-axis are missing. The images of the scratch wounds are so small that the reader can't see any differences. What should be shown with the photos? Unclear to me.

Fig. 6: Also here the photos are too small.

References

There are some typos in the References, especially in the upper and lower case of titles. For example, MTA and SEM are always capitalized. Some references are incomplete, for example No. 15 and 16.

The order of the sections in the manuscript is not correct. According to the General Considerations of the authors guidelines of “Materials”, the sequence of the sections of a research manuscript  should be: Introduction, Results, Discussion, Materials and Methods, Conclusions.

There are some typos, the manuscript must be proofread in depth.

Overall, the results are confirmatory.

Author Response

First of all we would like to thank the editor and the referees for the interest they have shown in our article entitled "Comparative cytocompatibility and mineralization potential of Bio-C Sealer and TotalFill BC Sealer." (materials-594755).

We wish to thank the editor and the referees for their constructive comments and suggestions. Please find attached the new version which it has been reviewed in line with your comments and those of the referees.

As it is mentioned in your letter, changes are highlighted in the new manuscript, using the "track changes" mode in MS Word format.

Reply to Reviewer #2:

Concern of the reviewer: Unfortunately, the current literature on the subject has not been thoroughly reviewed. Some studies, that have investigated the effect of calcium silicate-based sealers on PDL cells, are not cited. Our response: Thank you for the valuable suggestions provided. We expect to be able to fulfill your concerns in the new manuscript; the changes are highlighted. Concern of the reviewer: “Classically, ideal endodontic sealers should be bio-inert, i.e., they do not interact with the host tissues to form a mineralized interface [2].“ doesn't make any sense to me. Why shouldn't sealers induce mineralization? Our response: Thank you; to avoid confusions, we deleted this sentence. Concern of the reviewer: Material and Methods. In this study, the cells were only contacted with eluates from hard-set sealer. However, this does not correspond to a clinical scenario where freshly mixed, unset sealer can be pressed over the apex during root canal filling and come into direct contact with the cells ("puff"). This may lead to different results than indicated here. Why were these dilutions chosen? Why no other dilutions? Could this have led to different results? This point needs to be discussed.

Our response: Thank you for your comments. Recently have been published a paper about the setting time of Bio-C Sealer and Total Fill (Zordan-Bronzel et al. 2019, JOE). The setting time of these sealers were 4 and 10 hours, respectively. So, in the clinical situation, the significant effect of these sealers is in set form. The preparation of dilutions was performed according to previous studies and according to the International Organization for Standardization (ISO) guideline 10993-12 and the ratio of the specimen surface area was 1.5 cm2/mL (ISO 10993-5)[15]. For this reason, we used undiluted, 1/2, ¼ dilutions.

Concern of the reviewer: The study also investigated the mineralization performance of the cells. Why were osteoblasts, which are responsible for bone mineralisation, not tested?

Our response: Thank you very much for your comments. As this referee commented in concern 1, previous authors have bee investigated the effect of calcium silicate-based sealers on PDL cells. Stem cells from periodontal tissues are the first cells in contact with calcium silicate-based sealers and then, osteoblasts. Also, these cells can form cementum and bone. So, hPDLSCs have mineralization capacity.

Concern of the reviewer: In the present study it set AH Plus was cytotoxic to the hPDLSCs. This is in contrast to other studies where it was shown that after setting, AH Plus was no longer cytotoxic [Eldeniz et al. 2007, Zhou et al. 2015, Silva et al. 2016, Jung et al. 2019]. This point needs to be discuss

Our response: Thank you very much for your comments. In our opinion, there is evidence that calcium silicate-based root canal sealers have excellent biological properties compared to conventional resin-based sealer in set form. We added more references in the Discussion section.

Jung S, Sielker S, Hanisch MR, Libricht V, Schäfer E, Dammaschke T. Cytotoxic effects of four different root canal sealers on human osteoblasts. PLoS One. 2018 Mar 26;13(3):e0194467. Seo DG, Lee D, Kim YM, Song D, Kim SY. Biocompatibility and Mineralization Activity of Three Calcium Silicate-Based Root Canal Sealers Compared to Conventional Resin-Based Sealer in Human Dental Pulp Stem Cells. Materials (Basel). 2019 Aug 5;12(15).

Lee BN, Hong JU, Kim SM, Jang JH, Chang HS, Hwang YC, Hwang IN, Oh WM. Anti-inflammatory and Osteogenic Effects of Calcium Silicate-based Root Canal Sealers. J Endod. 2019 Jan;45(1):73-78.

Concern of the reviewer: Page 3, line 95: what does "OD" mean? Please expand abbreviations. Our response: Thank you for your appreciation. This sentence is now corrected in the manuscript. Concern of the reviewer: Please, “Carl Zeiss“ without “Inc.“, TotalFill without “®“. Our response: We appreciate this consideration; this is now corrected in the manuscript. Concern of the reviewer: The discussion suddenly breaks off and came to a sudden end. The methods are not clearly discussed and current literature is not cited. Please, avoid the use of the possessive pronoun “we“. Our response: We accepted the reviewer suggestion, and we have been modified the discussion section. Concer of the reviewer: 1: The image resolution in Fig. 1 is very low. The graphs are therefore difficult to decipher. Fig. 2: Why are the photos of the scratch wounds and the graphs in an image? That makes little sense to me. The units of the y-axis are missing. The images of the scratch wounds are so small that the reader can't see any differences. What should be shown with the photos? Unclear to me. Fig. 6: Also here the photos are too small.

Our response: Thank you very much. We have modified the figures accordingly.

Concern of the reviewer: There are some typos in the References, especially in the upper and lower case of titles. For example, MTA and SEM are always capitalized. Some references are incomplete, for example No. 15 and 16.

Our response: Changed performed. Once again, thank you very much for all your valuable comments.

Concern of the reviewer: The order of the sections in the manuscript is not correct. According to the General Considerations of the authors guidelines of “Materials”, the sequence of the sections of a research manuscript should be: Introduction, Results, Discussion, Materials and Methods, Conclusions.

Our response: Thank you very much. We comproved that the order is optional See: https://www.mdpi.com/journal/materials/instructions#preparation. There are a lot of papers in Materials with the standard order: Introduction, Materials and Methods, Results, Discussion, Conclusions. For example:

Edrees HY, Abu Zeid STH, Atta HM, AlQriqri MA. Induction of Osteogenic Differentiation of Mesenchymal Stem Cells by Bioceramic Root Repair Material. Materials (Basel). 2019 Jul 19;12(14). pii: E2311. Alamoudi RA, Abu Zeid ST. Effect of Irrigants on the Push-Out Bond Strength of Two Bioceramic Root Repair Materials. Materials (Basel). 2019 Jun 14;12(12). pii: E1921. Gambarini G, Miccoli G, Seracchiani M, Khrenova T, Donfrancesco O, D'Angelo M, Galli M, Di Nardo D, Testarelli L. Role of the Flat-Designed Surface in Improving the Cyclic Fatigue Resistance of Endodontic NiTi Rotary Instruments. Materials (Basel). 2019 Aug 8;12(16). pii: E2523.

Concern of the reviewer: There are some typos, the manuscript must be proofread in depth. Our response: Thank you for your comments. Changed performed.

Round 2

Reviewer 1 Report

The authors have addressed some of the issues.

Reviewer 2 Report

After revision the manuscript is clearly improved now. But there are still some typos.